# Three New Species of *Absidia* (Mucoromycota) from China Based on Phylogeny, Morphology and Physiology

**Heng Zhao** [1,2], **Yong Nie** [3], **Tongkai Zong** [4], **Yucheng Dai** [1,*] **and Xiaoyong Liu** [2,*]

1 Institute of Microbiology, School of Ecology and Nature Conservation, Beijing Forestry University, Beijing 100083, China; zhaoheng181@mails.ucas.ac.cn
2 College of Life Sciences, Shandong Normal University, Jinan 250014, China
3 School of Civil Engineering and Architecture, Anhui University of Technology, Ma'anshan 243002, China; lyly19851207@126.com
4 Key Laboratory for Forest Resources Conservation and Utilization in the Southwest Mountains of China, Ministry of Education, Southwest Forestry University, Kunming 650224, China; zongfungi@163.com
* Correspondence: yuchengdai@bjfu.edu.cn (Y.D.); liuxiaoyong@im.ac.cn (X.L.)

**Abstract:** Species of *Absidia* are distributed widely in the environment, while their diversity is insufficiently studied. Three new species, *A. frigida*, *A. gemella* and *A. longissima*, are proposed herein from Xinjiang and Yunnan in China based on phylogenetic, morphological and physiological evidence. According to maximum likelihood (ML), maximum parsimony (MP) and Bayesian inference (BI) analyses, the phylogenetical results suggest that *A. frigida*, *A. gemella* and *A. longissima* are closely related to *A. psychrophilia*, *A. turgida* and *A. zonata* and *A. koreana*, respectively, based on ITS and LSU rDNA sequences. *Absidia frigida* is characterized by a lower growth temperature, which does not grow above 24 °C. It differs from *A. psychrophilia* by sporangiophores, sporangia, columellae, collars and projections. *Absidia gemella* is distinguished from *A. turgida* by hypha, sporangiospores, sporangia, projections and sporangiophores. *Absidia longissima* is discriminated from *A. zonata* and *A. koreana* by sporangiophores, columellae and collars. The three new species are described and illustrated in this article.

**Keywords:** basal fungi; new taxa; molecular phylogeny; fungal diversity

## 1. Introduction

*Absidia* Tiegh. is a core genus of early-diverging fungi, belonging to Mucoromycota, Mucoromycotina, Mucoromycetes, Mucoromycetes, Mucorales and Cunninghamellaceae (www.indexfungorum.org, accessed on 27 December 2021). *Absidia* was erected in 1876 and typified by *A. reflexa* Tiegh. [1,2]. Members of *Absidia* are widely distributed in soils, plant residues, herbivorous dung, decaying substrates and air [2–10]. In particular, some species are found in the mycangia of ambrosia beetles (*A. psychrophilia*) [11,12], as well as the body surface of bats (*A. stercoraria*: https://bccm.belspo.be/content/remarkable-fungal-biodiversity-northern-belgium-bats, accessed on 27 December 2021). Species within *Absidia* possess important metabolites for industrial and medical applications, such as steroids, α-galactosidase, laccase, fatty acids and chitosan [13–18].

Species of *Absidia* produce stolons. Sporangiophores form on the middle part of the stolons, while rhizoids form at both ends. Sporangia are multi-spored, pyriform to globose, deliquescent-walled and apophysate [2–4,19]. Columellae are conical, suglobose to globose or applanate, commonly with one to several projections [2,4,20,21]. Zygospores are contained in zygosporangia, and their opposite suspensory cells are appendaged [13,20].

Some mycologists and taxonomists, such as Bainier, Hagemann, Lendner, Hesseltine, Ellis and Schipper, not only described and classified the genus *Absidia* s.l. but also inferred its phylogeny based on morphological and physiological features [13,22–25]. Incorporating molecular data, morphological traits and growth temperatures, the traditional classification of *Absidia* s.l. has been revised and divided into three genera, namely,

*Lentamyces* (parasitic on other mucoralean fungi, optimum growth temperatures between 14 °C and 25 °C), *Absidia* s.s. (mesophilic, optimum growth temperatures between 25 °C and 34 °C) and *Lichtheimia* (thermotolerant, optimum growth temperatures between 37 °C and 45 °C) [13,20,21].

Currently, 43 species are described in *Absidia* (Table 1, www.indexfungorum.org, accessed on 13 December 2021), with type strains originating from 17 countries, including Australia, Brazil, Canada, China, Cuba, the Czech Republic, Egypt, France, Holland, India, Mexico, Pakistan, South Korea, Switzerland, Tanzania, Thailand and the USA. Until now, 14 species have been recorded in China [3,4,26], and more potential novel species are being illustrated [27]. In this paper, three new species, *A. frigida*, *A. gemella* and *A. longissima*, are described from soil in China according to phylogenetic, morphological and physiological evidence.

**Table 1.** The information of type strains of species in *Absidia*.

| Name | Authors | Record in China | Year of Publication | Host | Location of Type Strains |
|---|---|---|---|---|---|
| *Absidia reflexa* | Tiegh. | no | 1878 | unknown | France |
| *A. repens* | Tiegh. | yes | 1878 | unknown | France |
| *A. dubia* | Bainier | no | 1882 | unknown | Unknown |
| *A. caerulea* | Bainier | no | 1889 | on wet bread | France |
| *A. spinosa* | Lendn | yes | 1907 | soil | Switzerland |
| *A. cylindrospora* | Hagem | yes | 1908 | unknown | Unknown |
| *A. glauca* | Hagem | yes | 1908 | unknown | Unknown |
| *A. heterospora* | Y. Ling | yes | 1930 | pine forest soil | Holland |
| *A. fusca* | Linnem | no | 1936 | unknown | Unknown |
| *A. egyptiaca* | R. Sartory, J. Mey. & Tawfic | no | 1939 | unknown | Egypt |
| *A. cuneospora* | G.F. Orr & Plunkett | no | 1959 | soil | USA |
| *A. pseudocylindrospora* | Hesselt. & J.J. Ellis | yes | 1962 | soil | Tanzania |
| *A. psychrophilia* | Hesselt. & J.J. Ellis | yes | 1964 | glands of *Curculionidae* | Canada |
| *A. anomala* | Hesselt. & J.J. Ellis | no | 1964 | soil | Cuba |
| *A. californica* | J.J. Ellis & Hesselt | no | 1965 | dung of *Rattus* | USA |
| *A. clavata* | B.S. Mehrotra & Nand | no | 1967 | dung of *Bos taurus* | India |
| *A. macrospora* | Váňová | no | 1968 | mountain forest soil | Czech Republic |
| *A. fassatiae* | Váňová | no | 1971 | soil | Czech Republic |
| *A. inflata* | J.H. Mirza, S.M. Khan, S. Begum & Shagufta | no | 1979 | soil | Pakistan |
| *A. narayanae* | Subrahm. | no | 1990 | on bat guano | India |
| *A. idahoensis* | Hesselt., M.K. Mahoney & S.W. Peterson | yes | 1990 | *Nomia melanderi* | USA |
| *A. caatingaensis* | D.X. Lima & A.L. Santiago | no | 2015 | soil | Brazil |
| *A. koreana* | Hyang B. Lee, Hye W. Lee & T.T. Nguyen | no | 2015 | soil | South Korea |
| *A. stercoraria* | Hyang B. Lee, H.S. Lee & T.T.T. Nguyen | no | 2016 | dung of rat | South Korea |
| *A. panacisoli* | T. Yuan Zhang, Ying Yu, He Zhu, S.Z. Yang, T.M. Yang, Meng Y. Zhang & Yi X. Zhang | yes | 2018 | rhizosphere of *Panax notoginseng* | China |
| *A. terrestris* | Rosas de Paz, Dania García, Guarro, Cano & Stchigel | no | 2018 | soil | Mexico |
| *A. jindoensis* | Hyang B. Lee & T.T.T. Nguyen | no | 2018 | rhizosphere soil of *Coniferae* | South Korea |
| *A. cornuta* | D.X. Lima, C.A. de Souza, H.B. Lee & A.L. Santiago | no | 2020 | soil | Brazil |
| *A. pernambucoensis* | D.X. Lima, Souza-Motta & A.L. Santiago | no | 2020 | soil | Brazil |

**Table 1.** *Cont.*

| Name | Authors | Record in China | Year of Publication | Host | Location of Type Strains |
|---|---|---|---|---|---|
| *A. multispora* | T.R.L. Cordeiro, D.X Lima, Hyang B. Lee & A.L. Santiago | no | 2020 | soil | Brazil |
| *A. saloaensis* | T.R.L. Cordeiro, D.X Lima, Hyang B. Lee & A.L. Santiago | no | 2020 | soil | Brazil |
| *A. pararepens* | Jurjević, M. Kolařík & Hubka | no | 2020 | air | USA |
| *A. healeyae* | A.S. Urquhart & A. Idnurm | no | 2021 | leaf litter | Australia |
| *A. aguabelensis* | J.D. Leitão, T.R.L. Cordeiro, Hyang B. Lee & A.L. Santiago | no | 2021 | soil | Brazil |
| *A. montepascoalis* | L.W.S. Freitas, Hyang B. Lee, T.T.T. Nguyen | no | 2021 | soil | Brazil |
| *A. bonitoensis* | C.L. Lima, D.X. Lima, Hyang B. Lee & A.L. Santiago | no | 2021 | soil | Brazil |
| *A. ovalispora* | H. Zhao & X.Y. Liu | yes | 2021 | soil | China |
| *A. globospora* | T.K. Zong & X.Y. Liu | yes | 2021 | soil | China |
| *A. medulla* | T.K. Zong & X.Y. Liu | yes | 2021 | soil | China |
| *A. turgida* | T.K. Zong & X.Y. Liu | yes | 2021 | soil | China |
| *A. zonata* | T.K. Zong & X.Y. Liu | yes | 2021 | soil | China |
| *A. edaphica* | V.G. Hurdeal, E. Gentekaki, Hyang B. Lee & K.D. Hyde | no | 2021 | soil | Thailand |
| *A. soli* | V.G. Hurdeal, E. Gentekaki, Hyang B. Lee & K.D. Hyde | no | 2021 | soil | Thailand |
| ***A. frigida* \*** | **H. Zhao, Y.C. Dai & X.Y. Liu** | **yes** | **2021** | **soil** | **China** |
| ***A. gemella*** | **H. Zhao, Y.C. Dai & X.Y. Liu** | **yes** | **2021** | **soil** | **China** |
| ***A. longissima*** | **H. Zhao, Y.C. Dai & X.Y. Liu** | **yes** | **2021** | **soil** | **China** |

\* Species proposed herein are shown in bold.

## 2. Materials and Methods

### 2.1. Sample Collection and Strain Isolation

Soil samples were collected from Yunnan and Xinjiang in China in September 2021. Then, strains were isolated according to the method in previous studies [3,4]. In brief, soil (1 g) was suspended in sterile water (100 mL), and then the suspension (100 uL) was spread on plates with potato dextrose agar (PDA: 200 g potato, 20 g glucose, 20 g agar and 1000 mL distilled water) supplied with streptomycin sulfate (100 mg/mL) and ampicillin (100 mg/mL). The plates were incubated in the dark at 20 °C and 25 °C. Colonies were purified and then deposited in the China General Microbiological Culture Collection Center, Beijing, China (CGMCC). Cultures were also dried and deposited in the Herbarium Mycologicum Academiae Sinicae, Beijing, China (HMAS).

### 2.2. Morphology and Maximum Growth Temperature

Morphological observations and maximum growth temperature tests followed the method by Zheng et al. [28–33]. In brief, malt extract agar medium [34] (MEA: malt extract 30 g, peptone 3 g, agar 20 g, 1000 mL distilled water), a stereomicroscope (SMZ1500, Nikon, Tokyo, Japan) and a microscope (Axio Imager A2, Carl Zeiss, Oberkochen, Germany) were used. For describing morphological characteristics, a range between the minimum and maximum sizes based on a statistic of more than 20 measurements was adopted. For maximum growth temperature tests, plates were firstly incubated at 20 °C for 2 d, and then the incubation temperature increased by a gradient of 1 °C until the colonies stopped growing.

### 2.3. DNA Extraction, Amplification and Sequencing

Colonies were grown at 20 °C and 25 °C on synthetic mucor agar medium (SMA: dextrose 20 g, asparagine 2 g, $KH_2PO_4$ 0.5 g, $MgSO_4 \cdot H_2O$ 0.25 g, thiamin chloride 0.5 mg, agar 20 g, 1000 mL distilled water, pH7) for a week. Total cell DNAs were extracted with

a reagent kit (GO-GPLF-400, GeneOnBio Corporation, Changchun, China). A fragment covering the entire internal transcribed spacer (ITS) and a partial large subunit of ribosomal DNA (LSU rDNA) were amplified with the primer pair NS5M and LR5M (5′-GGC TTA ATT TGA CTC AAC ACG G-3′ and 5′-GCT ATC CTG AGG GAA ACT TCG-3′, respectively). The polymerase chain reaction (PCR) program followed Zhao et al. [3]. Sanger sequencing for PCR products was conducted by an external company (BGI Tech Solutions Beijing Liuhe Co., Limited, Beijing, China). ITS and LSU rDNA sequences were assembled and proofread with Geneious 9.0.2 (http://www.geneious.com, accessed on 1 May 2021) and then submitted to GenBank under the accession numbers in Table 2; top hits of the BLAST search for ITS sequences are provided in Supplementary Table S1.

**Table 2.** The GenBank accession numbers of sequences used in this study.

| Species * | Strains ** | GenBank Accession nos. | |
| --- | --- | --- | --- |
| | | ITS | LSU |
| *Absidia anomala* | CBS 125.68 [T] | NR_103626 | NG_058562 |
| *A. bonitoensis* | URM 7889 [T] | MN977786 | MN977805 |
| *A.caatinguensis* | URM 7156 [T] | KT308169 | KT308171 |
| *A. californica* | CBS 314.78 | MH861141 | MH872902 |
| *A. californica* | FSU 4747 | AY944872 | EU736300 |
| *A. californica* | FSU 4748 | AY944873 | EU736301 |
| *A. coerulea* | CBS 101.36 | MH855718 | MH867230 |
| *A. coerulea* | FSU 767 | AY944870 | AF113443 |
| *A. cornuta* | URM 6100 | MN625256 | MN625255 |
| *A. cuneospora* | CBS 101.59 [T] | NR_159602 | NG058559 |
| *A. cylindrospora* | CBS 100.08 | JN205822 | JN206588 |
| *A. edaphica* | MFLU 20–0416 | MT396372 | MT393987 |
| **A. frigida** | **CGMCC 3.16201** [T] | **OM108487** | **OM030223** |
| *A. fusca* | CBS 102.35 | NR103625 | NG058552 |
| **A. gemella** | **CGMCC 3.16202** [T] | **OM108488** | **OM030224** |
| *A. glauca* | CBS 129233 | MH865253 | MH876693 |
| *A. glauca* | CBS 101.08 [T] | NR_111658 | NG_058550 |
| *A. glauca* | FSU 660 | AY944879 | EU736302 |
| *A. globospora* | CGMCC 3.16031 [T] | MW671537 | MW671544 |
| *A. globospora* | CGMCC 3.16035 | MW671538 | MW671545 |
| *A. globospora* | CGMCC 3.16036 | MW671539 | MW671546 |
| *A. heterospora* | SHTH021 | JN942683 | JN982936 |
| *A. jindoensis* | CNUFC-PTI1-1 [T] | MF926622 | MF926616 |
| *A. koreana* | EML-IFS45-1 [T] | KR030062 | KR030056 |
| **A. longissima** | **CGMCC 3.16203** [T] | **OM108489** | **OM030225** |
| *A. macrospora* | FSU 4746 | AY944882 | EU736303 |
| *A. medulla* | CGMCC 3.16034 [T] | MW671542 | MW671549 |
| *A. medulla* | CGMCC 3.16037 | MW671543 | MW671550 |
| *A. multispora* | URM 8210 [T] | MN953780 | MN953782 |
| *A. ovalispora* | CGMCC 3.16018 [T] | MW264071 | MW264130 |
| *A. panacisoli* | SYPF 7183 [T] | MF522181 | MF522180 |
| *A. pararepens* | CCF 6352 | MT193669 | MT192308 |
| *A. pernambucoensis* | URM 7219 [T] | MN635568 | MN635569 |
| *A. pseudocylindrospora* | CBS 100.62 [T] | NR_145276 | NG_058561 |
| *A. pseudocylindrospora* | EML-FSDY6-2 | KU923817 | KU923814 |
| *A. psychrophilia* | FSU 4745 | AY944874 | EU736306 |
| *A. repens* | CBS 115583 [T] | NR103624 | HM849706 |
| *A. saloaensis* | URM 8209 [T] | MN953781 | MN953783 |
| *A. soli* | MFLU 20-0414 | MT396373 | MT393988 |
| *A. spinosa* | FSU 551 | AY944887 | EU736307 |
| *A. stercoraria* | EML-DG8-1 [T] | NR_148090 | KT921998 |
| *A. terrestris* | FMR 14989 [T] | LT795003 | LT795005 |
| *A. turgida* | CGMCC 3.16032 [T] | MW671540 | MW671547 |

**Table 2.** *Cont.*

| Species * | Strains ** | GenBank Accession nos. | |
| | | ITS | LSU |
| --- | --- | --- | --- |
| *A. zonata* | CGMCC 3.16033 [T] | MW671541 | MW671548 |
| *Cunninghamella blakesleeana* | CBS 782.68 | JN205869 | MH870950 |
| *C. elegans* | CBS 167.53 | JN205882 | HM849700 |

* Sequences obtained herein are shown in bold. ** The "[T]" represents type strains.

### 2.4. Phylogenetic Analyses

For reconstructing phylogenetic trees, all the sequences were aligned with AliView (version 3.0) [35] and MAFFT (version 7, https://mafft.cbrc.jp/alignment/server/, accessed on 12 December 2021), and then manual proofreading was performed. Maximum likelihood (ML), maximum parsimony (MP) and Bayesian inference (BI) analyses were all adopted for phylogenetic analyses as described in Nie et al. [36,37], using RAxML (version 8) with the GTRGAMMA substitution model [38], PAUP (version 4.0b10) [39] and MrBayes (version 3.2.7a) [40], respectively. Finally, sequence alignments and phylogenetic trees were deposited in TreeBase (submission ID SS29123).

## 3. Results

### 3.1. Taxonomy

In this study, we propose three new species of *Absidia* from Xinjiang and Yunnan, China (Tables 1 and 2; Figures 1–4). All these novel taxa were demonstrated by molecular sequences, morphology and physiology.

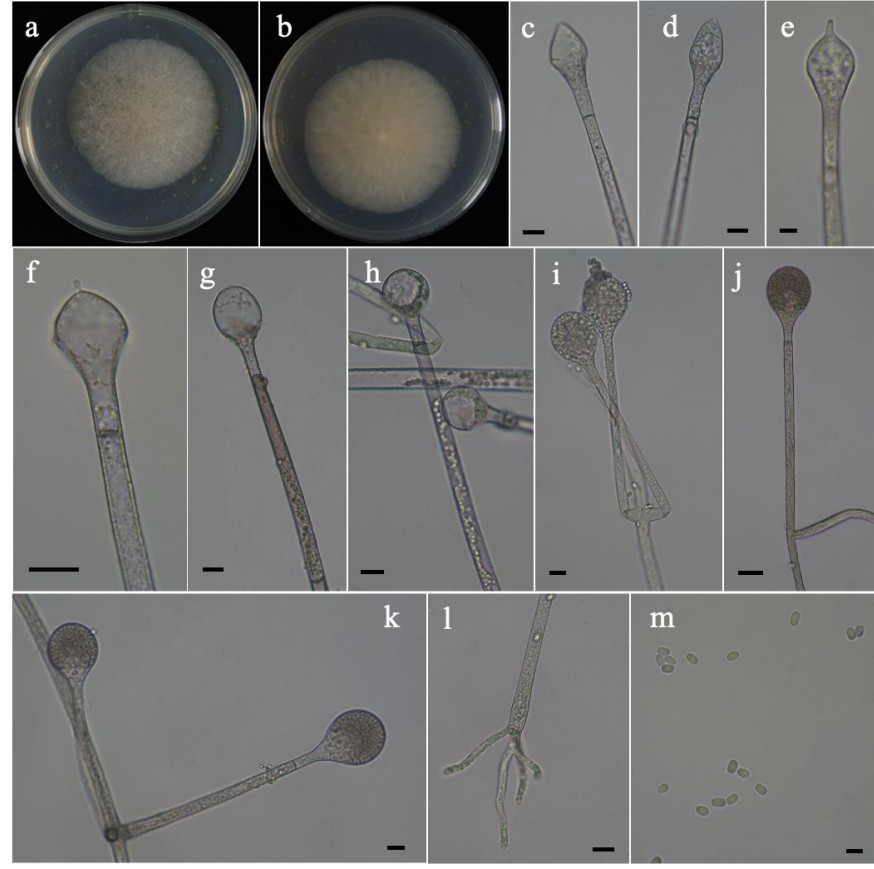

**Figure 1.** Morphologies of *Absidia frigida* ex-holotype CGMCC 3.16201. (**a**,**b**) Colonies on MEA, (**a**) obverse, (**b**) reverse; (**c**–**h**) columellae and projection; (**i**–**k**) sporangium; (**l**) rhizoids; (**m**) sporangiospores. Scale bars: (**c**–**l**) 10 μm, (**m**) 5 μm.

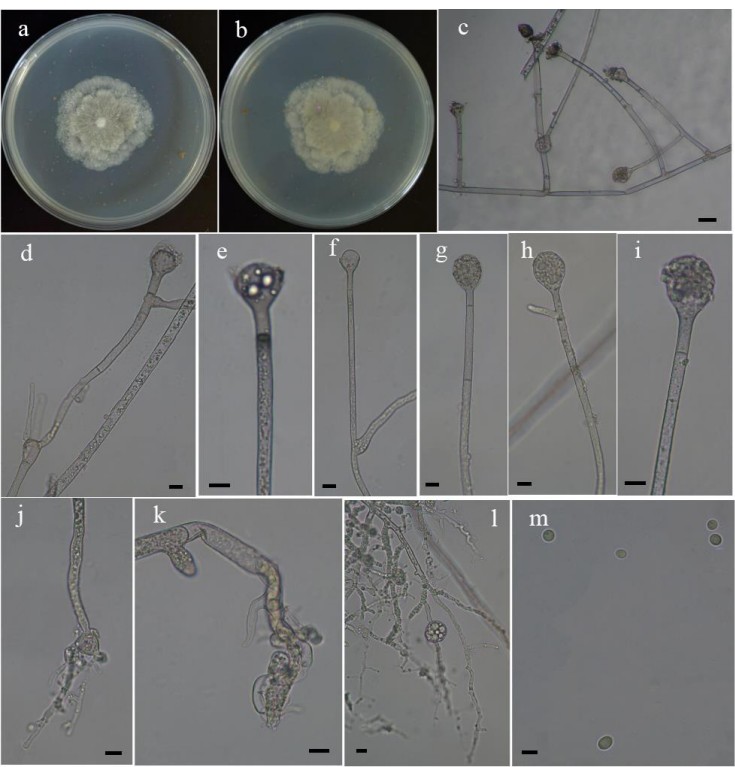

**Figure 2.** Morphologies of *Absidia gemella* ex-holotype CGMCC 3.16202. (**a**,**b**) Colonies on MEA, (**a**) obverse, (**b**) reverse; (**c**) monopodial sporangiophores; (**d**) sympodial sporangiophore; (**e**,**f**) columellae; (**g**–**i**) sporangium; (**j**,**k**) rhizoids; (**l**) substrate hyphae with swellings; (**m**) sporangiospores. Scale bars: (**c**) 20 μm, (**d**–**l**) 10 μm, (**m**) 5 μm.

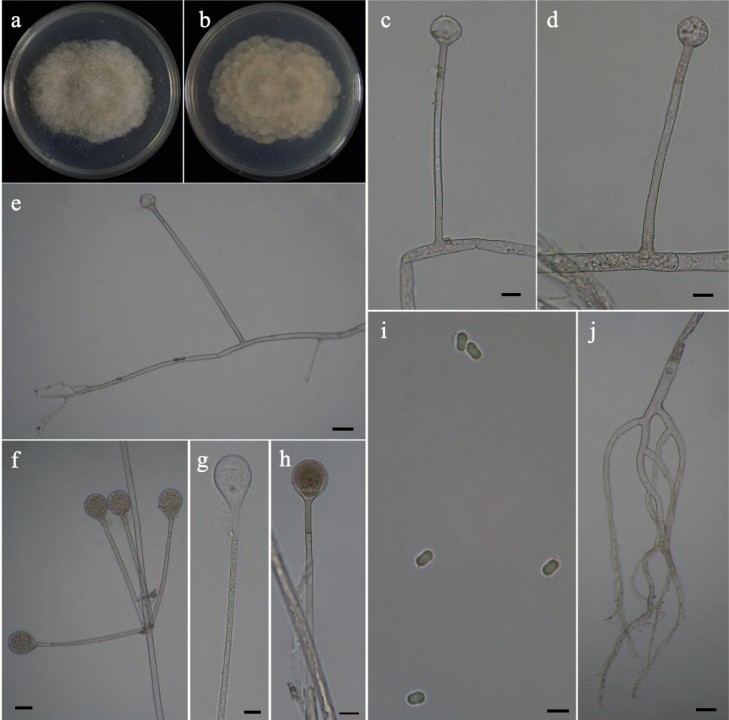

**Figure 3.** Morphologies of *Absidia longissima* ex-holotype CGMCC 3.16203. (**a**,**b**) Colonies on MEA, (**a**) obverse, (**b**) reverse; (**c**), (**d**) columellae and projection; (**e**) sporangiophores arising from stolons borne on rhizoids; (**f**–**h**) sporangium or sporangia; (**i**) sporangiospores; (**j**) rhizoids. Scale bars: (**c**,**d**,**g**,**h**) 10 μm, (**e**,**f**,**j**) 20 μm, (**i**) 5 μm.

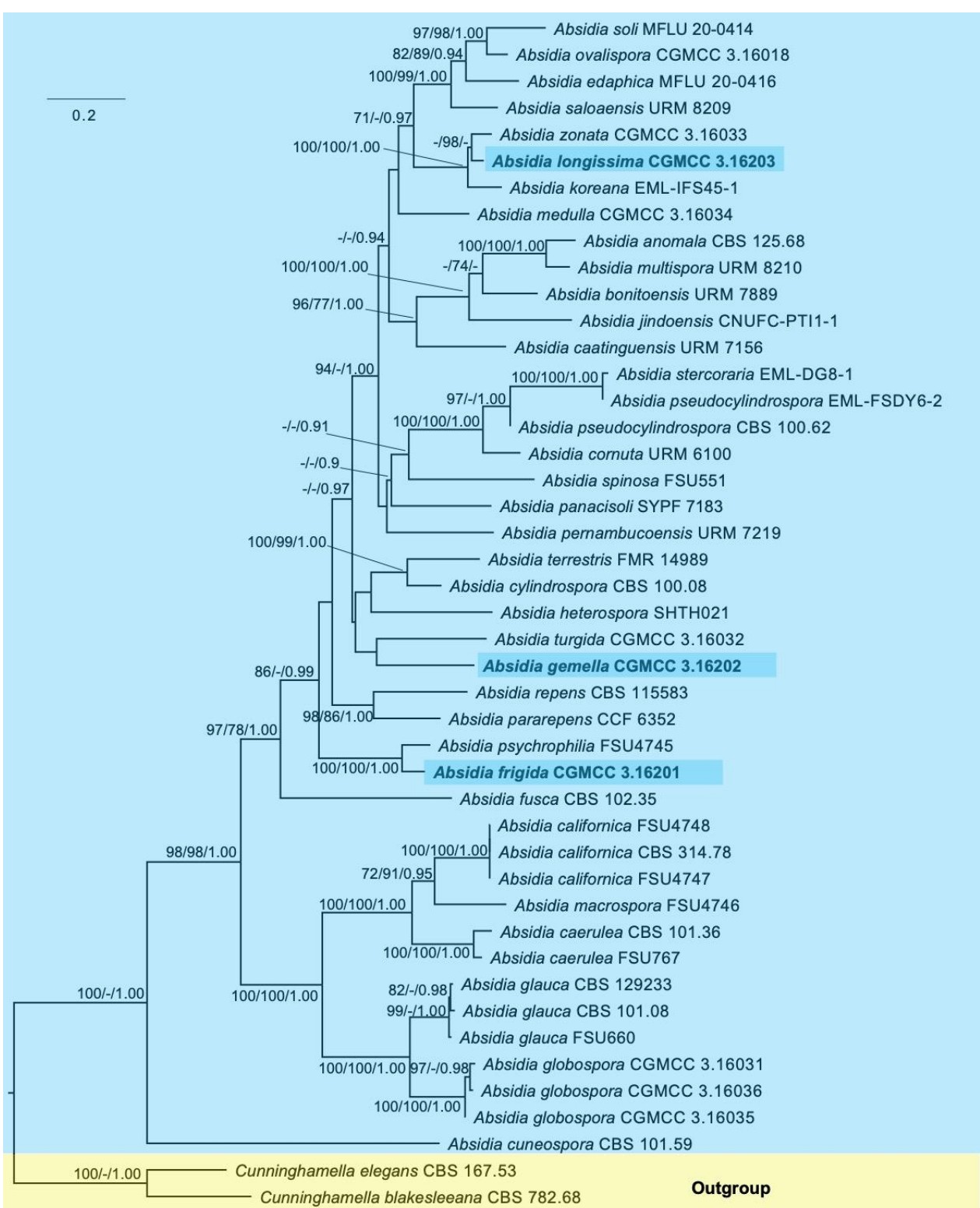

**Figure 4.** Maximum likelihood phylogenetic tree of *Absidia* based on ITS and LSU rDNA sequences, with *Cunninghamella elegans* and *C. blakesleeana* as outgroups. The three new species, *A. frigida*, *A. gemella* and *A. longissimi*, are shaded. Maximum likelihood (ML) bootstrap values (≥70%)/maximum parsimony (MP) bootstrap values (≥70%)/Bayesian inference (BI) posterior probabilities (≥0.9) of each clade are indicated along branches. Scale bar in the upper left indicates substitutions per site.

3.1.1. *Absidia frigida* H. Zhao, Y.C. Dai & X.Y. Liu, sp. nov.

*Fungal Names*: FN570961. Figure 1.

*Etymology*: *frigida* (Lat.) referring to a relatively lower maximum growth temperature of no more than 24 °C.

*Holotype*: HMAS 351587.

*Description*: Colonies on MEA at 20 °C for 7 days, growing moderately slow, attaining 60 mm diameter, white at first, gradually becoming light brown, regularly at reverse. Hyphae branched, hyaline at first, sometimes brownish when mature, aseptate when juvenile, septate with age, 5.5–18.5 μm wide. Stolons branched, hyaline, brownish, smooth, with septa. Rhizoids finger-like, rarely present, always unbranched, rarely branched, few swollen at the top. Sporangiophores arising from rhizoids, or substrate hyphae, erect or slightly bent, 1–4 in whorls, unbranched or simple branched, while sympodial not observed, hyaline, sometimes with a septum 16.0–23.0 μm below apophyses, 21.0–300.0 μm or more in length and 3.0–7.0 μm in width. Sporangia subglobose to pyriform, deliquescent-walled, smooth, multi-spored, colorless when young, pigmented when elder, 13.0–33.0 μm long and 12.5–32.0 μm wide. Apophyses distinct, hyaline, subhyaline or slightly pigmented, 4.5–9.5 μm high, 3.5–7.0 μm wide at the base and 7.0–11.5 μm wide at the top. Collars absent. Columellae conical, ovoid, elliptical, subglobose to globose, hyaline or subhyaline, smooth, 16.0–23.0 μm long and 15.5–18.0 μm wide, or 13.0–20.5 μm in diameter. Projections present or absent, if present, always one only, rarely two, small, hyaline or subhyaline, 1.5–3.0 μm long. Sporangiospores cylindrical, hyaline to subhyaline, smooth, 2.5–4.5 μm long and 2.0–3.0 μm wide. Zygospores not observed. Chlamydospores absent.

*Maximum growth temperature*: 24 °C.

*Material examined*: China, Xinjiang, Ili Kazak, Zhaosu County, 43°13′58″ N, 81°10′45″ E, altitude: 2219 m, from soil sample, 31 October 2021, Heng Zhao (holotype HMAS 351587, living ex-holotype culture CGMCC 3.16201).

3.1.2. *Absidia gemella* H. Zhao, Y.C. Dai & X.Y. Liu, sp. nov.

*Fungal Names*: FN570962. Figure 2.

*Etymology*: *gemella* (Lat.) referring to producing two types of sporangiospores.

*Holotype*: HMAS 351588.

*Description*: Colonies on MEA at 27 °C for 7 days, growing slow, attaining 45 mm diameter, irregular zonate, white at first, become brown when older, irregular at reverse. Hyphae branched, hyaline at first, brownish when mature, aseptate when juvenile, septate with age, 4.5–17.0 μm wide, substrate hyphae well developed, always branched, sometimes formed elliptical, subglobose to globose swollen in the internal. Stolons branched, hyaline, brownish, smooth, septate. Rhizoids root-like, not well developed, always branched, rarely with a septum at the top. Sporangiophores arising from stolons, erect or slightly bent, often monopodial, simple branched, commonly sympodial, rarely 2–5 in whorls, sometimes a swelling beneath sporangium, hyaline, with one to several septa, 38.0 to more than 350.0 μm and 3.5–6.0 μm in width. Sporangia oval to pyriform, deliquescent-walled, smooth, multi-spored, colorless when young, pigmented when old, 16.0–25.5 μm long and 16.5–24.0 μm wide. Apophyses distinct, subhyaline to hyaline, 3.0–7.5 μm high, 3.5–7.0 μm wide at the base and 7.0–14.0 μm wide at the top. Collars present, small. Columellae globose, subglobose to elliptical, hyaline or pigmented, smooth or rough, 11.0–22.0 μm long and 12.5–17.5 μm wide. Projections absent. Sporangiospores two types, cylindrical, or subglobose to globose, hyaline or subhyaline, smooth, 3.0–4.5 μm long and 2.5–4.0 μm wide, or 2.5–3.5 μm in diameter. Zygospores not observed. Chlamydospores absent.

Maximum growth temperature: 29 °C.

*Material examined*: China, Xinjiang, Altay City, Burqin County, 28°37′1″ N, 87°2′58″ E, altitude: 1330 m, from soil sample, 31 October 2021, Heng Zhao (holotype HMAS 351588, living ex-holotype culture CGMCC 3.16202).

3.1.3. *Absidia longissima* H. Zhao, Y.C. Dai & X.Y. Liu, sp. nov.

*Fungal Names*: FN570963. Figure 3.

*Etymology*: *longissima* (Lat.) referring to producing very long rhizoids.

*Holotype*: HMAS 351589.

*Description*: Colonies on MEA at 27 °C for 7 days, growing moderately fast, attaining 70 mm diameter, irregular concentrically zonate with ring, white at first, gradually becoming gray, irregular at reverse. Hyphae branched, hyaline at first, brownish when mature, aseptate when juvenile, septate with age, 3.5–11.5 μm wide. Stolons branched, hyaline, brownish, smooth, with septa. Rhizoids root-like, well developed, always branched. Sporangiophores arising from stolons, erect or slightly bent, 2–5 in whorls, monopodial, unbranched or simple branched, while sympodial not observed, hyaline, with a septum 12.5–19.5 μm below apophyses, 40.0–190.0 μm or more in length and 2.5–4.5 μm in width. Sporangia globose to pyriform, deliquescent-walled, smooth, multi-spored, colorless when young, brownish when old, 9.5–28.0 μm long and 10.0–28.0 μm wide. Apophyses small, sometimes not distinct, subhyaline to hyaline, slightly pigmented, 2.5–7.5 μm high, 3.0–6.0 μm wide at the base and 4.5–17.5 μm wide at the top. Collars absent. Columellae subglobose to globose, hyaline, smooth, 7.0–11.5 μm in diameter. Projections present or absent, if present, one only. Sporangiospores cylindrical, hyaline, smooth, slightly contracts in center, 3.0–4.0 μm long and 1.5–2.0 μm wide. Zygospores not observed. Chlamydospores absent.

*Maximum growth temperature*: 36 °C.

*Material examined*: China, Yunnan Province, Qujing, Huize County, from soil sample, 15 November 2021, Heng Zhao (holotype HMAS 351589, living ex-holotype culture CGMCC 3.16203).

*3.2. Phylogenetic Analyses*

The phylogenetic trees of individual ITS and LSU rDNA are provided in Supplementary Figure S1a,b, respectively. The concatenated ITS and LSU rDNA sequence dataset consists of 45 taxa, including 36 species of *Absidia* and 2 species of the outgroup *Cunninghamella*. A total of 1652 sites are composed of 619 constant, 775 parsimony-informative and 258 parsimony-uninformative characters. The maximum parsimony (MP) tree result shows that the tree length (TL), consistency index (CI), homoplasy index (HI), retention index (RI) and rescaled consistency index (RC) are 5600, 0.3493, 0.6507, 0.5326 and 0.1860, respectively. The best model of Bayesian inference (BI) is GTR + I + G, and the average standard deviation of split frequencies is no more than 0.01. The topology of the maximum likelihood tree (ML) was chosen to represent the phylogenetic relationship (Figure 4), since ML, MP and BI resulted in similar topologies. The results suggest that *Absidia frigida* is closely related to *A. psychrophilia* (100/100/1.00); *Absidia gemella* is next to *A. turgida*; and, finally, *A. longissima* is closely related to *A. zonata* and *A. koreana* (100/100/1.00).

**4. Discussion**

The ITS and LSU rDNA tree (Figure 4) shows the phylogenetic positions of the three new species in the genus *Absidia*. In detail, *A. frigida* is closely related to *A. psychrophilia* with sufficient support values (100/100/1.00). However, *A. frigida* physiologically differs from *A. psychrophilia* by maximum growth temperature (25 °C vs. 28 °C) [11]. Additionally, morphologically, *A. frigida* is distinguished from *A. psychrophilia* by less whorls of sporangiophores (four vs. eight), smaller sporangia (13.0–33.0 μm long and 12.5–32.0 μm wide vs. 20.0–50.0 μm in diameter), smaller columellae (16.0–23.0 μm long and 15.5–18.0 μm wide, or 13.0–20.5 μm in diameter vs. 6.5–30.0 μm in diameter), no collars (absent vs. indistinct) and smaller projections (1.5–3.0 μm vs. 6.5 in length) [11].

*Absidia gemella* clusters with *A. turgida*, though their sibling relationship does not obtain a strong support. Physiologically, the maximum growth temperature of *A. gemella* is lower than that of *A. turgida* (30 °C vs. 33 °C). Morphologically, *A. gemella* is differentiated from *A. turgida* by narrower hyphae (4.5–17.0 μm wide vs. 9.0–23.0 μm wide),

narrower sporangiospores (3.5–6.0 μm in width vs. 4.5–11.0 μm in width), different shape of sporangia (oval to pyriform vs. globose to pyriform), smaller sporangia (16.0–25.5 μm long and 16.5–24.0 μm wide vs. 20.5–42.5 μm long and 20.0–41.5(–46.0) μm wide) and no projections (absent vs. 9.5 μm in length) [4]. Moreover, sporangiophores in *A. gemella* are monopodial, simple branched, commonly sympodial and rarely 2–5 in whorls, while in *A. turgida*, sporangiophores are 1–4 in whorls, unbranched or sometimes simple [4].

Phylogenetically, *Absidia longissima* is most closely related to *A. zonata* (-/98/-) and *A. koreana* (100/100/1.00). Physiologically, *A. longissima*, *A. zonata* and *A. koreana* also possess similar maximum growth temperatures: no growth at 37 °C, no growth at 38 °C and restricted growth at 37 °C, respectively [4,41]. However, *A. longissima* morphologically differs from *A. zonata* and *A. koreana* by whorls of sporangiophores (2–5 in whorls vs. 2–5(–8) in whorls vs. 2–6 in whorls), shape of columellae (subglobose to globose vs. hemispherical vs. globose), smaller columellae (7.0–11.5 μm in diameter vs. 9.5–19.0 μm long and (6.0–)7.5–14.5(–16.5) μm wide vs. 11.4–19.0 μm long and 11.0–17.0 μm wide) and collars (absent vs. absent or present vs. present around each columella) [4,41].

Morphologically, sporangiospores and projections play essential roles in distinguishing *Absidia* from other genera [13,20,21]. However, as the numbers of species in *Absidia* gradually increased, some species were found to possess two or more different shapes of sporangiospores, such as *A. gemella*, *A. multispora*, *A. pararepens*, *A. repens* and *A. turgida* [4,5,10], which suggested more characters should be adopted. In addition, the majority of species form projections on the apex of columellae, except *A. heterospora* [11]. The new species *A. gemella* does not produce projections on MEA plates even after ten days. Therefore, we believe that the morphological delimitation of the genus *Absidia* should be revised, especially for the character of projections.

Since 2015, a total of 22 species of *Absidia* have been proposed, outnumbering those described in the last century (Table 1). Remarkably, in the past two years, seven and five new species have been described in Brazil and China, respectively. At present, *Absidia* is already the second largest genus in the phylum Mucormycota (www.indexfungorum.org, accessed on 27 December 2021). Taking into account the 3 new species reported herein, 46 species are accepted from all around the world, among which 16 are recorded in China.

Currently, an increasing number of studies focus on fungal diversity and ecological distribution [3,4,42–48], which provides a foundation to deeply understand the kingdom Fungi. Although a total of 46 species are described in *Absidia*, their ecological distribution has still not been unraveled. A few of these species were found in dung, insects, leaf litter, etc., while most species were collected from soil (Table 1), which suggested that they may have complex ecological habits or hosts. In this study, the maximum growth temperature of the new species *A. frigida* was 24 °C, implying that some species of *Absidia* have adapted to low-temperature environments, and that maybe more potential species will be hidden in extreme habitats.

**Supplementary Materials:** The following are available online at https://www.mdpi.com/article/10.3390/d14020132/s1, Figure S1: Maximum likelihood phylogenetic tree of *Absidia* based on ITS and LSU rDNA sequences, respectively, with *Cunninghamella elegans* and *C. blakesleeana* as outgroup, Table S1: Top hits for the new species based on BLAST search for ITS sequences from type materials.

**Author Contributions:** Data curation, H.Z.; formal analysis, Y.D.; funding acquisition, Y.D. and X.L.; investigation, H.Z.; methodology, Y.N. and T.Z.; project administration, X.L.; software, Y.N. and T.Z.; writing—original draft, H.Z.; writing—review and editing, Y.D. and X.L. All authors have read and agreed to the published version of the manuscript.

**Funding:** This research was supported by the Second Tibetan Plateau Scientific Expedition and Research Program (STEP), Grant No. 2019QZKK0503, and the National Natural Science Foundation of China, Grant Nos. 31970009 and 32170012.

**Institutional Review Board Statement:** Not applicable.

**Data Availability Statement:** Sequences and trees have been deposited in GenBank (Table 2) and TreeBase (number: SS29123), respectively.

**Acknowledgments:** Many scholars are thanked for their help with sampling and depositing, and they are Bao-Kai Cui, Shun Liu, Chang-Ge Song and Tai-Min Xu (Beijing Forestry University); Ze-Fen Yu and Min Qiao (Yunnan University); and Ke Wang, Zhuo Du and You-Zhi Wang (Institute of Microbiology, Chinese Academy of Sciences).

**Conflicts of Interest:** All authors declare no conflict of interest.

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
