# Peer review of "Three New Species of Absidia (Mucoromycota) from China Based on Phylogeny, Morphology and Physiology"

_diversity, doi:10.3390/d14020132_

Round 1

Reviewer 1 Report

Please carefully check the figure legends, 

Author Response

Response to reviewer 1

  1. Repeat

Respond: The two “Tiegh.” are applied to the genus and species separately, so we prefer to keep them but move to the beginning of this paragraph where the genus first appears.

  1. j has only one

Respond: Thanks. We use “Sporangium” instead of “Sporangia”.

  1. Only have one Sporangium in each

Respond: Thanks. We use “Sporangium” instead of “Sporangia”.

  1. not in the figure

Respond: Thanks for this suggestion. The figure 3 legend has been updated as follows:

Figure 3. Morphologies of Absidia longissima ex-holotype CGMCC 3.16203. a, b. Colonies on MEA (a, obverse, b, reverse); c, d. Collumellae and projection; e. Sporangiophores arising from stolons borne on rhizoids; f–h. Sporangium or sporangia; i. Sporangiospores; j. Rhizoids — Scale bars: c, d, g. h. 10 μm, e, f, j. 20 μm, i. 5 μm.

  1. Are these measurements correct?

Respond: Yes, these measurements are correct. The rhizoids of Absidia longissima are generally more than 200 μm long, which is unique in the genus Absidia.

  1. old fashion

Respond: Yes, we present phylogenetic results the old way.

  1. round up

Respond: Thanks. The numbers have been rounded.

Reviewer 2 Report

In this study, the Authors describe three new Absidia species isolated from Xinjiang and Yunnan in China based on phylogenetic, morphological, and physiological characters. As the overall Mucoromycota diversity is insufficiently studied, all records of new specimens are valuable. In this context, the presented work is an important contribution to increasing our knowledge. However, it needs several significant refinements to better support proposed conclusions.

My main concern is about describing new species based on one isolate. According to phylogenetic species definition, species is an evolutionarily divergent lineage (lineage means group of individuals),  that has maintained its hereditary integrity through time and space. It is the smallest group of populations that can be distinguished by a unique set of morphological or genetic traits. Having only one individual doesn’t allow to verify the intra- and interspecies variability (neither on genetic nor on morphological level). Studying one isolate and comparing it with another strain allows only to say that we have two different strains but it doesn’t allow us to draw conclusions on species delimitation. I recommend Authors to read the following guidelines: https://imafungus.biomedcentral.com/articles/10.1186/s43008-021-00063-1 by Aime et al. 2021 and address this issue in their paper.

I also attached pdf file with some more detailed comments.

Author Response

Response to reviewer 2

Dear Sir/Madam,

Thank you for your valuable suggestions. Please allow me to explain firstly the major issue about the representative number of a species. Traditionally, for a very long time, most new species of Absidia have been described and published with type strain only. We isolated these strains from environmental soil and then carefully purified them. Actually, we obtained more strains from the same soil sample but all resulted in the same morphology and DNA sequence. Therefore, we just selected one strain as material. We amplified out a clean PCR band with each primer pair and obtained a clean DNA sequencing for each locus. For robustness, we have indeed repeated these tests three times. Moreover, the top hits after BLAST of ITS sequences are summarized in Supplement Table 1. All these should be sufficient to guarantee their novelty.

More comments:

  1. I'd recommend to add info on type material availability here.

Respond: Thank you. Following your suggestion, we used the letter “T” to label the type material in the Table 2.

  1. In the description you refer to the temperature as to the main distinctive character. However, I miss information how you determined them. It would be advisable to add one paragraph in methods section on min max growth temperature experiments.

Respond: Thank you for your suggestion. We have put relative information in the revised manuscript: “In brief, plates firstly incubated at 20 °C for 2 d, and then the incubation temperature increased by a gradient of 1 °C until the colonies not growing”.

  1. AliView is one of programs but it doesn't determine the algorithm which was used. Was it mafft, muscle, one of clustals? This is crucial step of huge impact on final results.

Respond: Thank you. In order to verify the result, the AliView (version 3.0) and MAFFT (version 7) were used, and then manual proofreading was performed. We have revised the manuscript accordingly.

  1. what about substitution model used?

Respond: The GTRGAMMA substitution model was used for RAxML. We have revised the manuscript accordingly.

  1. italics

Respond: Thanks. We have italicized the “Absidia frigida” in the revised edition.

  1. what are these numbers? Is it min and max size (out of how many measurements)? Median or average? SD should be given... And here and after

Respond: Traditionally, taxonomy of zygomycetes describe morphological characteristics by using a range between the min and max sizes based on a statistic of more than 20 measurements. We followed this tradition.

  1. italics

Respond: Thanks. We have italicized the “Absidia gemella” in the revised edition.

  1. The order of described structures was different in previous taxon. It would be easier to compare them if you keep the same order of structures.

Respond: Thank you. We accepted your advice.

  1. It means you lack molecular data for 6 already described taxa. Right? Could you comment on that?

Respond: Right. We do not used these molecular data of the six taxa because they are not closely related to the three new species we described in this paper.

  1. To say that something forms group you should have more than one element.

Respond: Thanks. The described have been corrected as “A. frigida is closely related to A. psychrophilia with well support values (100 / 100 / 1.00).”

  1. I lack information on morphological comparison with taxa described but lacking molecular data.

Respond: We summarized the BLAST top hits for the three new species based on ITS sequences in Supplementary Table 1.

  1. add reference

Respond: Thanks. The data was originated from the database of the Index Fungorum, and we have added the website in the revised paper.

Reviewer 3 Report

Comment 1: What are the top hits after you run BLAST of your ITS sequences? You can include them in the supplementary information so the reader can judge the novelty of these three species based on some quantitative measurement.

Comment 2: It would be interesting to label the type strains in the table 2 by letter T.

Author Response

Response to reviewer 3

Comment 1: What are the top hits after you run BLAST of your ITS sequences? You can include them in the supplementary information so the reader can judge the novelty of these three species based on some quantitative measurement.

Respond: Thanks, that’s a good idea, we summarized the top hits based on BLAST search for ITS sequences from type material in the Supplementary Table 1.

Comment 2: It would be interesting to label the type strains in the table 2 by letter T.

Respond: Thank you. We followed your suggestion and used the letter “T” to label the type strains in the Table 2.

Round 2

Reviewer 2 Report

The Authors improved the manuscript and clarified several issues. I only don't feel convinced by the argument that others are also using one strain or not giving standard deviation. If the Authors are saying that min-max values are given after measurements of 20 structures, it should be clearly explained in the methods section. There were different schools of performing measurements between different groups of mycologists in the past. I can't see the reason why measurements of zygomycetes should be performed in a different way than measurements of any other fungal group. BLAST hit is not enough for me as an argument for something to be new species. However, I think that the paper can be published in its current form.

Author Response

Dear Sir or Madam,

Thank for your advice. We added the method for measurements in the section of Morphology and maximum growth temperature, “For describing morphological characteristics, a range between the min and max sizes based on a statistic of more than 20 measurements was adopted.” Moreover, three new species of genus Absidia were illustrated by phylogenetic, morphological and physiological evidence, while BLAST hit only.

Best wishes!

Heng Zhao